The role of past experience in development of feeding behavior in common vampire bats

Berrío-Martínez Jineth 1
Kaiser Samuel 1
Nowak Michelle 1
Page Rachel A. 1
Carter Gerald G. carter.1640@osu.edu gcarter1640@gmail.com 1 2
1 Smithsonian Tropical Research Institute , Balboa , Ancón , Panama
2 Department of Evolution, Ecology and Organismal Biology, The Ohio State University , Columbus , OH , United States of America
Curley James
Electronic publication date: 2019 Aug 5
Publication date: 2019
Volume: 7
Electronic Location ID: e7448
Received 2019 Mar 12; Accepted 2019 Jul 10
Copyright: ©2019 Berrío-Martínez et al.
Copyright year: 2019
Copyright holder: Berrío-Martínez et al.
License: This is an open access article distributed under the terms of the Creative Commons Attribution License, which permits unrestricted use, distribution, reproduction and adaptation in any medium and for any purpose provided that it is properly attributed. For attribution, the original author(s), title, publication source (PeerJ) and either DOI or URL of the article must be cited.
License URL: https://creativecommons.org/licenses/by/4.0/

Keywords: Learning, Bats, Vampire bats, Development, Captivity

Funding: Smithsonian Postdoctoral Fellowship Smithsonian Institution Scholarly Studies Grant: ‘Tracking and manipulating cooperative relationships in vampire bats’ Gerald Carter was supported by a Smithsonian Postdoctoral Fellowship. Work by Gerald G. Carter and Rachel A. Page was supported by Smithsonian Institution Scholarly Studies Grant: ‘Tracking and manipulating cooperative relationships in vampire bats’. The funders had no role in study design, data collection and analysis, decision to publish, or preparation of the manuscript.

==============================
The life history strategy of common vampire bats (Desmodus rotundus) suggests that learning might play a role in development of their foraging skills. We took advantage of 12 captive births in a study colony of vampire bats to test the role of past experience in two aspects of feeding. First, we compared preferences for blood temperature in 32 wild-born vampire bats versus 11 captive-born vampire bats that had only previously fed on blood of ambient temperature or colder. We found no evidence for a preference in either group for blood presented at 4 °C versus 37 °C. Second, we tested whether captive-born vampire bats with no previous experience of feeding on live animals could successfully feed on a live chicken. Five of 12 naïve captive-born bats were able to bite the chicken and draw blood, but only one bat gained more than 5% of body mass. We were unable to reasonably compare their feeding performance with that of wild-born bats because only two of three wild-born, short-term captive bats fed on the chicken and none of the seven wild-born, long-term captive mothers attempted to feed. This unexpected lack of feeding might be due to a previously reported age-dependent neophobia. When six of the captive-born bats were released in the wild, they appeared to feed successfully because they survived for more than three consecutive nights. We suggest further tests that would better clarify the role of learning in the development of foraging in vampire bats.

Introduction

Common vampire bats, Desmodus rotundus (E. Geoffroy, 1810), possess a suite of traits associated with complex foraging skills that are learned gradually over time (Schuppli, Isler & van Schaik, 2012). These traits include flexible foraging, slow development, individualized social relationships, extended parental care, and sharing of food with nonkin (Schuppli, Isler & van Schaik, 2012). Vampire bats are arguably flexible foragers because they rely on a combination of vision, echolocation, passive listening, olfaction, mechanoreception, and thermoperception to extract blood from a diverse range of vertebrate hosts (Greenhall, 1988; Schmidt et al., 1991; Gröger & Wiegrebe, 2006). They have a long lifespan, with records of 29 years in captivity (GGC, personal observation) and at least 17 years in the wild (Delpietro et al., 2017), and are an outlier among bats with regards to their maternal investment and slow reproduction. Gestation requires 5.5 to 7.3 months, and sexual maturity requires up to 9.5 months (Schmidt, 1988a; Delpietro & Russo, 2002). Juveniles are born large (roughly a quarter of the mother’s mass) and depend on their mother for longer than any other bat species; mothers nurse juveniles for 8 to 10 months. In contrast, lactation periods in temperate and other neotropical bats are typically three weeks to three months (Jenness & Studier, 1976; Kunz & Robson, 1995; Hamilton & Barclay, 1998; Crichton & Krutzsch, 2000; Delpietro & Russo, 2002; Kwiecinski, Falzone & Studier, 2003; Chaverri & Kunz, 2006). Female vampire bats form long-term social relationships that involve regurgitations of blood to unfed individuals in need (Wilkinson, 1984; Wilkinson, 1985; Delpietro & Russo, 2002; Carter & Wilkinson, 2013; Delpietro et al., 2017). This cooperative social support is especially important for younger bats because vampire bats will die after three consecutive unfed nights and individuals younger than two years fail to feed on roughly one-third of nights (Wilkinson, 1984). Vampire bats have one of the largest brain and neocortical volumes for their body size among bats (Baron, Stephan & Frahm, 1996; Bhatnagar, 2008). Finally, young vampire bats show increased exploration and decreased neophobia that peaks at 9-10 months (Park, 1990; Vrtilek et al., 2018). This unique combination of social complexity, slow life history, and flexible foraging make vampire bats a good candidate for a large role for learning in foraging. On the other hand, vampire bats are obligate blood-feeders and this extreme dietary specialization can also lead to evolutionary losses in learning ability. For instance, vampire bats have lost the widespread adaptive specialization of taste aversion learning (Ratcliffe, Fenton & Galef, 2003).

How important is learning during development of feeding behavior in vampire bats? To begin to answer this question, we took advantage of an existing captive colony of both captive-born and wild-born bats to test the role of past experience in feeding behavior. In the first test, we asked whether captive-born vampire bats show an innate preference for warm blood. Vampire bats possess unique heat receptors near their nostrils which allow them to detect blood-rich areas on the skin of their hosts at distances of up to 16 cm (Kürten & Schmidt, 1982; Schmidt & Manske, 1982). One previous study found that wild vampire bats will consume blood as cold as 14 °C (Bullard & Shumake, 1973), but the role of past experience in blood-temperature perception and preference remains ambiguous. We compared the blood-temperature preferences of wild-born bats with captive-born bats that had previously only fed on chilled or ambient temperature blood. If learning plays a key role in their use of thermoperception, then we expected that captive-born vampire bats would differ from wild-born bats. Specifically, wild-born bats should show a greater preference for warmer blood (37 °C) over artificially chilled blood (4 °C).

In the second test, we asked whether captive-born vampire bats could successfully feed from a live animal, given their complete lack of experience with this task. If a critical period of early experience or social learning is required for extracting blood from a live animal, then naïve captive-born bats, reared without experiencing feeding on live host animals, might be unable to feed successfully on live hosts. In the wild, younger vampire bats have been observed feeding from the same wound as their mothers (Wilkinson, 1985). Therefore, to test for whether the presence of mothers helps their offspring learn how to feed, we tested naïve captive-born bats alone and again in the presence of their wild-born mothers.

Materials & Methods

Animal subjects

As test subjects, we used 12 captive-born common vampire bats (Desmodus rotundus, six females, six males) and 24 wild-born long-term captive females. Additionally, we tested eight adult common vampire bats that were recently captured from the wild near Rio Indio, Chilibre in Chagres National Park, Panama. Bats were housed at the Smithsonian Tropical Research Institute (STRI) in Gamboa, Panama. Experiments were conducted from June to August 2017. We tested captive-born subjects aged from 6 to 16 months, which is after the period when captive young vampire bats begin feeding on blood on their own (Schmidt, 1988b). The long-term captive bats had fed on live animals before being captured in either December 2015 or June 2016. In captivity, they were fed with bovine or pig blood that was about 4 °C when first presented and that rose to ambient temperature (roughly 30 °C) during the night. The recently captured wild bats had no past experience with blood colder than the body temperature of their host. In contrast, the captive-born bats had no past experience with live prey or blood that was warmer than ambient temperature.

Research was approved by the STRI Animal Care and Use Committee (#2015-0915-2018-A9) and by the Panamanian Ministry of the Environment (#SE/A-76-16) and adhered to the standards of the American Society of Mammalogists Guidelines for use and handling of wildlife mammals for research (Sikes et al., 2016).

Experiment 1: Do captive-born vampire bats prefer warm blood?

Experimental procedure

We presented an unfed individual vampire bat in a 28 × 28 × 40 cm acrylic and mesh cage with two adjacent feeders, one filled with warm blood (37 °C) and the other filled with cold blood (4 °C). Each silo-style feeder was designed to provide water to birds and had one spout. We modified the feeders by insulating the tubes with polystyrene foam. The two spouts were positioned about 5 cm apart and were filled from the reservoir tubes of blood (∼20 mL). Cold blood was chilled in a refrigerator until the trial began and warm blood was warmed in a hot-water bath (Polyscience Waterbath, Niles, Illinois). Each feeder was then immediately placed into polystyrene foam insulators designed to maintain the temperature of the blood. The placement of feeders on the left or right was alternated for each trial. Trials lasted about one hour (mean = 62 min, range = 52 to 74 min), except for two interrupted trials that were 25 and 32 min respectively. We confirmed that the warmer blood always remained warmer during trials, by taking five measures of feeder-blood temperature over time. After one hour, the mean temperature of the insulated blood only changed by 5 °C, and the bats chose a feeder within the first 15 min in most trials (70%), with a median latency to choose of 2.8 min. Blood temperatures were therefore very different when bats chose a feeder.

We used an infrared surveillance camera to record the bat’s behavior and measure the amount of time a bat spent drinking from each blood spout. As a response variable, we defined feeding time bias as the difference in the proportion of time spent feeding on warm blood versus cold blood (e.g., 1.0 for 100% warm blood; 0 for 50% warm blood, and −1.0 for 100% cold blood). We initially measured the mass change in the two feeders by weighing the feeders immediately before and after the trial, but we found that this measure was inaccurate due to condensation forming on the colder tube, and a small amount of blood spilling due to the expansion of the colder air in the tube. For the best measure of consumption, we therefore used seconds of drinking time, which predicted the change in feeder weight (Pearson’s r = 0.77, df =  − 128, p < 0.0002). We also scored the bat’s first choice and the number of feeding events for each spout, where a feeding event is defined as the bat putting its mouth into the spout. Our conclusions do not differ when we instead compared differences in feeder weight or number of feeding events.

We tested 11 captive-born vampire bats and 24 long-term captive bats twice and tested the eight wild bats three or four times. We took the mean feeding time bias for each bat as our observational unit. To infer whether mean feeding time biases differed from zero, we used bootstrapping with 5000 samples (Canty & Ripley, 2015) to calculate the 95% confidence interval of the mean feeding time bias. These confidence intervals are a better alternative to post hoc power analysis for interpreting null results (Levine & Ensom, 2001).

Experiment 2: Can captive-born vampire bats feed on live prey?

Experimental procedure

For test subjects, we used 12 captive-born vampire bats described above, seven of their wild-born mothers, and three recently wild-captured adults. For live prey, we used 15 adult hens, Gallus gallus, because they are easy to handle and maintain, they pose little threat to the bats, and studies show that wild common vampire bats will readily feed on live chickens in proportion to their availability (e.g., Greenhall, Schmidt & Lopez-Forment, 1971; Greenhall, 1972; Bobrowiec, Lemes & Gribel, 2015). Hens were individually marked with colored leg bands, such that no hen was fed on more than once every 20 days. Hen health was carefully monitored. All hens remained healthy and active throughout the experiment. We saw no evidence of long-term repercussions from the bat feedings.

In each trial, we presented one hen to an unfed captive-born vampire bat in an experimental arena and left them undisturbed until sunrise. The experimental arena was 50 × 50 × 69 cm with three glass sides, and a mesh side and ceiling. Trials started between 2319 and 2346 h, except for one trial that started at 0059 h. During this time, we used an infrared surveillance camera system to record video from multiple angles. For the chicken, we provided a wooden perch elevated about 4 cm from the floor and a water dish. We used a fan to ventilate the experimental arena and to help maintain an ambient temperature. As a refuge for the vampire bat, we attached a mesh plastic tube covered with black plastic to a top corner of the arena. We weighed all vampire bat subjects before and after each trial. Before releasing the chicken the next morning, we photographed any bites on the chicken. The arena was cleaned between trials. We used the video recordings to score interactions among vampire bats and chickens, including: latency until feeding, duration of feeding events, number of attempted bites before a successful feeding, and duration of time spent moving around outside the refuge.

To test whether the presence of mothers would increase the feeding success of their young, we also tested five female and one male captive-born bats simultaneously with their mother, which were captured from the wild (near Tolé or Las Pavas, Panama) then housed in captivity for one to four months. We did not test reproductively mature sons with their mothers to avoid possible mating attempts and fighting. Trials with and without the presence of mothers were conducted in random order.

To compare with data from the captive-born bats, we also conducted six control trials with the wild-born bats. In the first three control trials, we tested an adult long-term captive mother alone. In the next three control trials, we individually tested two females and one male that were recently captured from the wild.

Results

Experiment 1. Do captive-born vampire bats prefer warmer blood?

We did not detect a blood temperature preference in captive-born or wild-born vampire bats (Fig. 1, Data S1). The bats appeared to detect both spouts, choose independently of their temperature, and simply consume more blood from whichever spout they chose first (Fisher’s exact test; odds ratio = 26, n = 41, p < 0.0001). The first choice of spout explained 62% of the variation in the feeding time bias, and the subject drank from only one spout in 71% of 135 trials.

Experiment 2. Can captive-born vampire bats feed on live prey?

We found that captive-born vampire bats could successfully bite a live chicken without previous experience or maternal assistance. Their feeding success, however, was ambiguous. Five of the 12 captive-born bats bit the chicken when tested alone, and one of the six captive-born bats bit in the presence of their mother (Table 1, Video S1), yet four of the five captive-born bats that fed did not gain more than 1 g of mass, showing that their feeding performance was poor (Table 2, Data S2). The oldest captive-born vampire bat (16 months old) fed on live prey to an extent that was comparable to the wild-born adults, and it gained a 12% increase in body mass (Table 2), but the other four captive-born bats that were capable of making a wound and feeding from it, did not gain much weight (−3% to +3% change in body mass).

Figure 1 No clear temperature preference detected in captive-born and wild-caught vampire bats.

The feeding time bias is the difference in the proportion of time spent feeding on warm blood (positive) versus cold blood (negative).

Discussion

Neither the captive-born nor wild-born vampire bats preferred warmer blood. Wild vampire bats will feed from wounds created by other vampire bats, so it is not too surprising that they readily drink from an open spout of blood (Greenhall, Schmidt & Lopez-Forment, 1971; Wilkinson, 1985). It is more surprising, however, that their reliance on thermoperception did appear to not generalize to this novel situation. Vampire bats rely on thermoperception to find blood vessels near the surface of the skin, and will choose to bite the warmer of two rabbit ears (Kürten & Schmidt, 1982; Schmidt & Manske, 1982). Yet a previous study reported that, when eight wild vampire bats were presented with choices of blood at 14 °C, 30 °C, 38.5 °C, and 47 °C, the bats did not show a temperature bias except for an avoidance of the 47 °C blood, possibly due to the denaturing of the blood proteins (Bullard & Shumake, 1973). Our study repeated this test using a more statistically powerful approach: we tested more subjects with a choice between two temperatures that were more different from each other. Our results corroborated the past findings and show that wild-born vampire bats will feed on blood as cold as 5 °C, even in the presence of blood at a more natural warm temperature. Clearly, heat is not the only cue that vampire bats use to select a blood source. The subjects presumably detected and chose spouts using other modalities such as olfaction and echolocation, and thermoperception is thus only one aspect of a vampire bat’s multimodal assessment of a bite site or blood source.

Table 1 Captive born bats fed on live chicken.

Categories of each type of bat that fed on live chickens.

Category of bat	N bats	Feeding in a trial	
Wild-born, short-term captive	3	2 of 3 trials when alone	
Wild-born, long-term captive	7	0 of 9 trials when alone, 0 of 6 trials with offspring	
Captive-born	12	5 of 12 trials when alone, 1 of 6 trials with mother	

Table 2 Feeding activity of vampire bats that did feed.

Birth	Age (months)	Sex	Body mass (g)	Drank water	Mass change (g)	Mass change (% mass)	Latency to feed (hours)	Time active(hours)	
captive	8	F	32.65	Yes	−1.08	−3%	2.3	7.9	
	11	M	29.27	No	−0.70	−2%	3.4	7.3	
	11	M	24.64	No	0.69	+3%	3.9	6.9	
	12a	F	32.04	Yes, No	0.51	+2%	3.1	7.2	
	16	M	26.44	Yes	3.28	+12%	4.0	6.4	
wild	>12	F	35.07	No	4.21	+12%	3.1	5.0	
	>12	M	25.87	Yes	4.83	+19%	1.0	5.6	
Notes.

a bat fed once alone and once with its mother (mean of the two similar values are shown).

Captive-born vampire bats could feed on a live animal (Table 1), but it is difficult to interpret their feeding performance (Table 2). Rapid urination causes large changes in body mass that can complicate the measurement of blood consumption in vampire bats (Wimsatt & Guerriere, 1962). Nevertheless, the bats in this test appeared to feed less than expected under typical circumstances. Even when only considering the bats that did gain mass, the average gain (∼3 g, Table 2) was still less than half of the mass gains observed in the wild (∼6 g, Turner, 1975) or in captivity (∼9 g, Breidenstein, 1982).

Two of the three wild-born, short-term captive vampire bats fed on the chicken successfully (Table 2). Unexpectedly, the adult long-term captive mothers did not feed on the chicken at all, either when tested alone (three trials), or with their daughter (six trials). We also never observed the adults feeding on the chicken in six pilot trials with alternative experimental setups. There are several possible explanations for why none of the mothers fed nor even seemed motivated to feed. Older captive bats may have been less motivated to feed due to differences in metabolic demands, but this seems unlikely given the susceptibility of vampire bats to fasting (Wilkinson, 1984; Freitas et al., 2013). The adult bats had been fasted in trials for another experiment to induce food sharing (following Carter & Wilkinson, 2013), so they might have habituated or learned that they would eventually be released back into the main cage due to their experiences of being repeatedly isolated overnight without food. However, the captive-born bats had similar experiences.

Perhaps the most likely explanation is that adult vampire bats are more neophobic and less exploratory than younger vampire bats. A previous study showed that the same captive-born bats were an order of magnitude more likely to explore a novel object compared to their adult groupmates (Carter et al., 2018). In the feeding trials of this study, the younger captive-born bats in the experimental arena were more active than the adults with regards to jumping, walking, sniffing, and interacting with the chickens (mean hours of activity for adults = 5.1 h, 95% CI [4.1–6.3] h, n = 10; for young captive-born bats = 7.2 h; 95% CI [6.9–7.5] h, n = 12; Video S2). When tested alone, all 12 captive-born young explored the floor of the experimental area, whereas only one of three mothers did. When tested in mother-daughter pairs, five of the six captive-born daughters and three of the six wild-born mothers moved to the floor. Most of the captive-born bats left the refuge a few minutes after the experimenter left, whereas most of the adult bats did not explore the floor of the test cage at all during the whole night. When tested with her daughter, one of the mothers did not even leave the refuge the entire night. These observations are consistent with past tests showing age-dependent exploration or neophobia (Carter et al., 2018).

An anecdotal observation of age-dependent boldness in a novel feeding context comes from a different captive colony (described in Carter & Wilkinson, 2013). In this colony, one of the authors (GGC) noticed that several of the ten captive-born younger bats would often fly and land on the author when he entered the flight cage, sometimes climbing along his back or up and down his legs. When he held still, one vampire bat attempted to bite his ear. The same behavior was reported by another animal caretaker. In sharp contrast, none of the 22 older adult captive-born bats in the group approached or landed on people entering the flight cage, and they instead typically flew away to a corner or remained vigilant and motionless.

Despite the apparent difference in performance, the actual biting behavior of captive-born bats was generally consistent with the wild-born, short-term captive adults, and with past descriptions of feeding behaviors where individuals took anywhere from a few minutes to 40 min to bite an animal (Greenhall, Schmidt & Lopez-Forment, 1971; Greenhall, 1972; Greenhall, 1988). The chickens were only bitten on the digits, ankle, and areas near the tail that lacked feathers. This is consistent with observations that vampire bats readily target unprotected areas of skin (Greenhall, 1988). To test for more subtle differences in feeding performance, one needs a larger behavioral sample from captive-born and wild-born bats tested under controlled conditions.

In conclusion, naïve young without any relevant past experience can feed on live prey, but it remains unclear how the feeding development and performance of captive-reared vampire bats compares with more experienced wild-reared adults. Our results suggest that social learning from mothers plays at most a supplementary role in the acquisition of the flexible extractive foraging skills of vampire bats.

Conclusions

In this study, we took advantage of a long-term captive colony to test some ideas about the development of feeding behaviors in common vampire bats. We had two unexpected results. First, we observed no temperature preference for blood in either captive-born, long-term captive, or recently wild-capture vampire bats. When vampire bats fed from spouts, they did not generalize their thermal preferences for bite sites on live animals. Thermal cues used during normal feeding are therefore not used in every feeding context. Future studies could incorporate trials where both captive-born and wild-born vampire bats are presented with bite sites on live animals that vary in temperature or other traits (following Schmidt & Manske, 1982).

Second, naïve captive-born common vampire bats can feed on live prey. Further studies could test how lack of experience influences feeding performance. One possible improvement to our study design would be to fully habituate all subjects to the feeding arena, by allowing isolated bats to feed from a dish of blood in the arena over several nights, before presenting a live animal.

The role of learning in other bat species and mammals is not easy to predict. For example, frog-eating bats (Trachops cirrhosus) feed by eavesdropping on the mating calls of several prey species, such as katydids and frogs, some of which are toxic (Page & Jones, 2016). These bats possess traits that suggest adaptation for hunting frogs, such as increased innervation in the part of the cochlea that allows for low-frequency hearing, allowing them to hear in the range of frog advertisement calls (Bruns, Burda & Ryan, 1989). Under such conditions, where predators have adaptations for specific kinds of prey and mistakes are costly, one would expect that preferences for specific frog calls would be fixed. More than a decade of work, however, shows the opposite trend: frog-eating bats are highly flexible, with the ability to learn both asocially and socially, rapidly acquiring and reversing associations between calls and prey quality (Page & Ryan, 2005; Page & Ryan, 2006; Jones et al., 2013; Patriquin et al., 2018). In contrast, other studies show that bats also possess largely innate heuristics under unexpected conditions. For example, naïve captive-born insectivorous bats treat any sufficiently large horizontal smooth surface as water, even a surface that they can fly beneath (Greif & Siemers, 2010). Naïve captive-born flower-visiting Glossophaga bats are strongly and innately attracted to dimethyl disulfide, a component of many neotropical bat-pollinated flowers (Helversen, Winkler & Bestmann, 2000; Carter, Ratcliffe & Galef, 2010). Yet, the performance of these bats when learning simple cue-based associations and generalizing them to new environments is surprisingly bad (Stich & Winter, 2006), and this is because both innate preferences for dimethyl disulfide and learned preferences for novel cues can be overshadowed by spatial memory (Thiele & Winter, 2005; Stich & Winter, 2006; Carter, Ratcliffe & Galef, 2010).

Another purpose of our study was to assess whether captive-born vampire bats would be likely to survive after being released into the wild. Captive-born animals are typically less likely to survive than wild-caught individuals (Stoinski et al., 2003), but there is much variation between species and studies (e.g., Stoinski et al., 2003; Piep et al., 2008; Benson-Amram et al., 2014; Rogers et al., 2016; Abu Baker et al., 2018; Yang et al., 2018). Very little is known about how well captive-reared bats can acquire the necessary skills for foraging in the wild (Courts, 1997; Constantine, 2003; Ruffell & Parsons, 2009; Serangeli et al., 2012). Some insectivorous bat species can apparently go from captive hand-feeding to successful wild foraging, whereas others must first learn how to forage on flying insects in a flight cage (Kelly et al., 2008; Kelly et al., 2012; Serangeli et al., 2012). Captive-reared animals develop a smaller hippocampus (LaDage et al., 2009; Tarr et al., 2009), and are likely to be cognitively and physiologically different in other ways. Six of the captive-born vampire bats from our study were released back into the wild and observed visually and tracked with automated proximity loggers (Ripperger et al., 2016 Ripperger et al., 2019; in prep) until they left the site. We confirmed that all these bats survived for at least three days in the wild and four of the six survived for at least five or six days, before departing the site. We observed no deaths. These observations suggest that the captive-born bats were also capable of feeding on wild prey.

Supplemental Information

Data S1 Raw data

Click here for additional data file.

We thank Ahana Aurora Fernandez, Sebastian Stockmaier, Darija Josic, Vanessa Pérez, and May Dixon for help with care of chickens. We thank the Smithsonian Tropical Research Institute for logistical support and staff of the Ministry of Agricultural Development (MIDA) in Colón, Panama, especially Rogelio Singh, for help with catching the wild vampire bats.

Additional Information and Declarations

Competing Interests

Author Contributions

Animal Ethics

Field Study Permissions

Data Availability

The authors declare there are no competing interests.

Jineth Berrío-Martínez, Samuel Kaiser and Gerald G. Carter conceived and designed the experiments, performed the experiments, analyzed the data, contributed reagents/materials/analysis tools, prepared figures and/or tables, authored or reviewed drafts of the paper, approved the final draft.

Michelle Nowak and Rachel A. Page contributed reagents/materials/analysis tools, authored or reviewed drafts of the paper, approved the final draft.

The following information was supplied relating to ethical approvals (i.e., approving body and any reference numbers):

Research was approved by the STRI Animal Care and Use Committee (#2015-0915-2018-A9).

The following information was supplied relating to field study approvals (i.e., approving body and any reference numbers):

Research was approved by the Panamanian Ministry of the Environment (#SE/A-76-16).

The following information was supplied regarding data availability:

Raw data and code are available in the Supplemental File.

Videos are available at Figshare:

Carter, Gerald G. (2019): Video S1. Captive-born vampire bat biting a live animal for the first time Supplementary video for Berrío-Martínez et al. (2019) PeerJ. figshare. Media. https://doi.org/10.6084/m9.figshare.8188580.v1.

Carter, Gerald G. (2019): Video S2. Captive-born vampire bats exploring a cage with a live animal. Supplementary video for Berrío-Martínez et al. (2019) PeerJ. figshare. Media. https://doi.org/10.6084/m9.figshare.8220857.v1.

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
