# Peer review of "The role of past experience in development of feeding behavior in common vampire bats"

_PeerJ, doi:10.7717/peerj.7448_

## Round 0.1 · original submission · Minor Revisions

Please address each of the comments made by both reviewer 1 and reviewer 2.

·

Basic reporting

No comment

Experimental design

No comment

Validity of the findings

No comment

Additional comments

This manuscript is very clearly written. The methods used are adequate to the question being asked, and the results are very clear. The discussion is grounded in the results obtained, with little to no speculation. Thus, I really have no major comments. I have a couple of suggestions and comments on some very minor issues.

The first suggestion is to move the idea mentioned in lines 94-98 to the discussion. I think you have a very solid foundation for establishing predictions about learning in vampire bats without the ideas included here; in fact, these do not seem well supported, but I believe would be more valuable in your discussion.

My second minor suggestion is to remind the reader what each experiment is trying to accomplish in materials and methods. For example, either include some hints in the subtitle "Experiment x" that reminds us what you are trying to accomplish, or else start the first paragraph with something like "To asses if captive-born bats show a preference for warm blood, we presented an unfed...".

Third comment: line 257 should be mentioning table 2, not table 1.

In line 304 you say that porcupines and armadillos are exotic, which they are not. Do you mean here that they are rare, or perhaps not the vampire's regular prey?

In line 318, I would remove the word "null", and just leave it as "We had two unexpected results".

I have a minor confusion about your test subjects for experiment 2. You mention in the section "Animal subjects" (material and methods) that you only tested animals aged 6 to 16 months of age. However, it is clear that is experiment 2 you use some adult females (which you mention in the methods section for experiment 2, no doubt). So perhaps to avoid confusion, in the section animal subjects you can clarify this to avoid confusion later on?

·

Basic reporting

The human evolution framing needs work. Please see my general comments.

Experimental design

No comment.

Validity of the findings

No comment.

Additional comments

Review by Teague O'Mara

Quick summary:
The paper is nicely written and easy to read, the experiments seem to test what the authors set out to test, data and analysis are solid, but the framing of study is a big reach and its strategy should be rethought (although it is never really discussed later, for what it's worth). Even though the results are a bit underwhelming, it's great to see more developmental work on bats and I think this work adds excellent information regarding the peculiarities of vampire bats.

General Comments:
The way that the authors frame the study in the context of human learning over simplifies a large and complex literature on the interactions among life history, brain size, social complexity and the ratchet effects of cumulative culture, and food complexity. The study by Schuppli and colleagues is their citation for all of human life history evolution (and associated learning), but this only touches on many of these traits. Schuppli, Isler, & van Schaik target why humans have a late age of skill competency and tests this via several theoretical constructs, often referred to as 'needing to learn' or 'energetic constraints' related to brain sizes and ecological risks posed to juveniles. Schuppli really targets the length of the immature period / juvenility and subadulthood (measured by age at first reproduction, AFR, versus the age of adult skill acquisition, ASC). While most mammals have relatively early ASC relative to AFR, mammals with complex foraging niches have ASC more similar to AFR, and humans have ASC much later than AFR. Schuppli's reasoning was that the synergistic effects of complex social relationships, allomaternal care, post-weaning provisioning, and the cooperative pooling of food all come together to allow humans to reproduce much earlier than their foraging ecology should allow. van Schaik shows in other work that it was this social cooperation and complexity the allows for the expansion of large brains of humans, particularly the neocortex (and to some extent falls in line with Robert Barton's earlier proposals).

Vampires arguably have the most highly specialized feeding ecology of bats, and there has been an enormous amount of adaptation in their morphology, physiology, and neurology that has facilitated this. The main issue here is whether the feeding behavior of vampires is a complex skill that needs a lot of time to be learned, and is perhaps facilitated through adult food sharing (Schuppli's Fig 2C). Vampires do have many characters that make them similar to primates, but making the life history comparison directly between vampire bats and humans is a bit bold -- vampire bat specialization and life history is more similar to aye-ayes or some of the haplorhine primates than humans (or maybe just a capital breeder since blood is a relative poor resource).

Maybe the best way to present this would be instead of going directly to a simplified view of humans, instead present the suite of socioecological characters that make vampires unusual and show in various taxa how these characters generally accompany complex foraging skills and extended developmental periods needed to learn those skills. You could present humans as a special case where skill acquisition is even further extended beyond what is expected (= AFR), but this is only possible because of social features like resource pooling, allomaternal care, etc. Then why vampires are an interesting case because they show some, but not all of these characters and you want to test the role of preferences and learning in their feeding. Or something to that extent.

It would be interesting to then revisit some of these life history and behavioral traits in your discussion for why you didn't really find learning (particularly their ASC/AFR relationship -- seems a tidy heuristic at the moment). This would go beyond the current 'it's complicated' approach to learning in bats presented in the discussion/conclusions and suggesting the need for better experiments and larger sample sizes. My feeling is that since there is so much specialization in these bats that the motivation at the root of their feeding is instinctual, but that they need to practice to get really good at it. You have a tough mix of results. The young bats attempt to feed more on the chicken, but are pretty terrible at it -- which speaks to the importance of practice / learning. But there was no social facilitation observed in the data, just that some young bats try to feed on a prey item that only 2 of 3 wild bats fed on. It would be useful to know a couple of other developmental landmarks relative to their long lactation period e.g., when adult incisors are fully erupted & functional, and when young fledge, dispersal age, to understand the necessity for learning or how long there really is for these bats to practice before they are independent

Some specific comments:
As a matter of preference, double spacing a manuscript for review makes it easier to read. People who study this type of thing suggest that double spacing decreases read time and lateral masking, reduces the number of fixations, and results in more accurate return sweeps during reading. Especially for reading on a computer screen.

24. What is significant weight?

148-149. I'm not clear on the feeder setup. Were there 2 spouts per temp reservoir that were 5 cm apart or were the 2 feeders separated by 5 cm? Also, were these lixit type spouts, or did the blood pool out for them to feed from?

335-336. I disagree. I would not say that Trachops is highly adapted (at least not in the Desmodus specialization sense). It is very similar to the rest of the gleaners. It has _additional_ low-frequency hearing and many physiological and behavioral redundancies for when it captures toxic prey. It's an extremely flexible predator whose perception still plays by most cognitive rules. We still don't know what proportion of its natural diet really comprises frogs since they also are keen katydid hunters.

342-344. Disulfides are also likely essential for social learning via odor, at least in mice and rats (e.g., Munger et al 2010 DOI 10.1016/j.cub.2010.06.021), and it seems to also be the case for what we found using displaced cues in Uroderma bilobatum.

---

## Round 0.2 · accepted · Accept

Thank you for attending to the comments of both reviewers. The paper is a very welcome addition to the literature.

·

Basic reporting

No comment

Experimental design

No comment

Validity of the findings

No comment

Additional comments

All my suggestions to the previous version of this manuscript have been addressed, and I have no further comments. In fact, the manuscript has improved significantly and the methods and results are much more straightforward. Great work!

·

Basic reporting

no comment

Experimental design

no comment

Validity of the findings

no comment

Additional comments

The authors have nicely addressed my concerns. Thank you for the thoughtful and clear responses to both reviewers.